# The Image of Sustainability in European Regions Considering the Social Sustainability Index

**Aniela Bălăcescu [1],*, Marian Zaharia [2], Rodica-Manuela Gogonea [3] and Genu Alexandru Căruntu [1]**

[1] Faculty of Economic Sciences, Constantin Brancusi University of Targu Jiu, 210185 Targu Jiu, Romania
[2] Faculty of Economic Sciences, Petroleum-Gas University of Ploiesti, 100680 Ploiesti, Romania
[3] Faculty of Cybernetics, Statistics and Economic Informatics, Bucharest University of Economic Studies, 010552 Bucharest, Romania
* Correspondence: anielabalacescu@gmail.com

**Abstract:** Evaluating and comparing the sustainable development of regions or countries is fundamental in the management of economic and social systems. From the multitude of tools and means for qualitative and quantitative assessment of the level of sustainable development, which ensure the comparability of the performances of each state, there is a set of indicators known as the Sustainable Society Index, originally developed by the Dutch Sustainable Society Foundation and later developed by TH Köln–University of Applied Sciences. Taking this into account, the objective of the undertaken research was to identify the stages and the similarities and disparities between the European states grouped into the four regions (East, North, South and West), as well as the positioning of Romania in this European context. The obtained results emphasize the fact that between the four European regions, as well as between their component states, there are both similarities and differences, especially in areas such as Well-balanced Society, Climate and Energy, and Economic Well-being regarding the values and implicitly the stage of development of sustainable societies. The article is intended to be a catalyst for discussions on understanding the causes which produce differences regarding the sustainability of European states in the context of the EUs commitment to the 2030 Agenda for sustainable development. Additional efforts are needed in the recovery and resilience process, especially in Eastern and Southern European countries.

**Keywords:** sustainable society index; Europe; Romania; cluster analysis; statistical test

## 1. Introduction

The concept of sustainability does not have a unanimously accepted definition, and there are many interpretations that address a wide range of transversal problems [1–5]. The multi-dimensionality of sustainability has generated difficulties in its measurement, and over the years, in an attempt to evaluate sustainability, different methods and indicators have been built, each of which presents advantages and disadvantages [6–10].

One of the useful tools, necessary on the one hand for evaluation and on the other hand for ensuring comparability between the states of the world regarding the degree of implementation of a sustainable society, is the Sustainable Society Index, abbreviated SSI [11–15]. This index was originally developed by the Dutch Sustainable Society Foundation and has been developed since 2019 by TH Köln–University of Applied Sciences [16]. If the SSI initially included 154 states/territories, as of 2019, the latest version of the SSI includes 231 states.

The SSI (THSSI, 2022) is based on the definitions and concepts formulated in the document Our Common Future (WCED, 1987), also known as the Brundtland Report, launched in 1987 [17].

Starting from this document, following the international meeting in Rio de Janeiro, in 1992, a comprehensive plan of action, known as Agenda 21 [18], resulted, with objectives leading to the sustainable development of the states of the world targeting the beginning of the 21st century. In summary, these objectives aimed at economic growth (the importance of finding methods to generate economic growth without harming the environment), environmental protection (the development of investments in green technologies, the development of ecological cities, water conservation, renewable energy sources) and social equality (the social well-being of people, poverty eradication, equal opportunities).

In 2015, the UN Summit on Sustainable Development resumed the objectives established by Agenda 21, formulating a set of 17 objectives contained in a new document Transforming our world: the 2030 Agenda for Sustainable Development [19].

During the 26th edition of the United Nations convention on climate change held in Glasgow between October and November 2021, the European Union renewed its commitment to combating climate change [20].

Europeans are increasingly demonstrating that they are aware of the challenges of climate change, resource misuse and environmental degradation, and the development of political programs demonstrates this fact, and many European countries are international leaders in sustainable consumption and production practices [21].

However, less favorable social and economic developments in some European states are increasing divergences in Europe. These contrasts have been permanently discussed, and the first concerns of Europeans regarding sustainable development on the three dimensions (social, economic and environmental) have been reflected since the year 2000 in the ambitious objectives of the Lisbon Agenda [22]. Currently, the implementation of the 2030 Agenda is monitored in order to achieve the objectives of sustainable development, and the progress made by the countries is integrated into the European Semester [23].

EU policies in combating divergences and promoting cohesion are transposed into the national legislations of the member states in such a way as to have a positive impact on the economic, social and environmental realities and provide the desired transformations in the direction of sustainability.

The latest evaluations of the European Commission [24] regarding sustainability in the member states show major challenges in achieving the objectives of sustainable development in Romania [25–27]. As a result, in this paper, special attention was paid to Romania, all the more reason since the demographic projections for this country represent a source of concern from the perspective of social sustainability, considering, in particular, the rapid aging process of the population [28] and the phenomenon of exodus recorded in the last 20 years [29].

The purpose of this study is to provide an aggregate perspective of the performance in the sustainability process through the lens of SSI at the level of Europe, in general, and Romania, in particular. The research question is: what is the European/national performance in the process of social, economic and environmental sustainability? Thus, the study has a mondo and macroeconomic orientation.

Achieving the objectives regarding sustainable development is based on the recovery and resilience measures adopted both at the EU and the member state level; however, it faces multiple challenges. The study is extremely useful in identifying existing gaps at the European level from the perspective of social sustainability performance, capturing the attention of political decision-makers as well as researchers.

The paper is organized into six sections. The introduction is the first section which briefly states the importance of the problem under investigation, followed by a review of recent literature on the social dimension of sustainability. This is followed by Section 3, dedicated to the data sources and characteristics of the SSI, then Section 4, with the research methodology. Section 5 presents the results and discussions regarding regional similarities and disparities across Europe. The sixth and final section outlines the main conclusions drawn from the research, with limitations and directions for future research.

## 2. Literature Review

The literature includes a wide range of articles, empirical studies and reports on sustainability. Considering the objective of the research, we focused our attention on those papers that mainly investigated the social dimension of sustainability and the causes that produce differences regarding the social sustainability of states without deviating from the principle of holism, which assumes interdependence between the three major dimensions of sustainability: social, economic and environmental. Social sustainability is integrally linked to results in the field of ecological economy and environmental protection. The information in this section is useful to improve the understanding of the concept of social sustainability, its influencing factors and the evaluation of its performance.

Although there is no clear definition of social sustainability, most debates converge toward the idea of people's well-being in the context of a healthy environment [30–34].

Eizenberg and Jabareen [35] suggest that in order to define and understand the concept of social sustainability, a holistic approach to risk in social aspects is necessary; therefore, four concepts, each referring to major social aspects, must be considered: equity, safety, eco-prosumption and sustainable urban forms.

All efforts aimed at social well-being must be subordinated to the principle of ensuring a balance between resources and needs in the conditions of a healthy environment without jeopardizing the well-being of future generations [36–40].

Climate changes constitute the biggest challenges in ensuring sustainability because it directly, quantitatively and qualitatively affects the biosphere. Thus, two million deaths caused by natural disasters were recorded in the period 1970–2019 globally [41].

Regarding the economic losses caused by climate change over the last 40 years, at the level of Europe, in the 32 EEA member countries, they were 450–520 billion EUR in 2020 [42].

The high demand for energy is determined on the one hand by population growth and comfort requirements and on the other hand by technological progress and can have a negative impact on the environment if sustainable consumption that ensures the use of renewable energy and waste minimization is not taken into account [43–45].

The use of renewable sources in the energy mix to reduce carbon dioxide emissions and maintain optimal air quality parameters is increasingly necessary to ensure sustainability [46].

The population is vulnerable to the water supply. Humanity's existence depends on the existence and quality of water, and concentrated efforts must be made in the direction of its saving and optimal management to ensure sustainable development [47–50]. The loss of access to drinking water can have a negative impact on a population's health and food availability [51–53].

Practicing pro-environment and pro-social oriented sustainable consumption [54] is possible, and its basis is formal or non-formal education [55].

Smart technologies are considered by many authors to be essential tools that can play essential roles in activating and stimulating economic, environmental and social sustainability [56–58].

Regarding the measurement of social sustainability, various indices have been developed over time, such as the Inclusive Wealth Index (its composition includes social indicators including health, education, employment) [59]; the Social Progress Index (measures social performance using 60 indicators) [60]; the Better Life Index (it relies on 11 indicators to measure social sustainability and it has an interactive tool - Your Better Life Index) [61]; and the Happy Planet Index (describes social component aspects such as well-being, inequality of outcome, life expectancy) [62].

The multi-dimensionality of the concept of social sustainability cannot be completely captured by any existing index, although each of them captures more or less the social aspects. In this sense, the most conclusive indices are currently considered to be the Sustainable Society Index [63] and the Sustainable Development Goals Index [64].

In conclusion, we state that there are multiple and different approaches to the phenomenon of social sustainability, which is natural considering the various interdependent relationships between people and the external ecosystem. The decision remains for the political, economic and social factors as to whether to continue exploiting the resources or to become involved in finding solutions to regenerate so that a sustainable ecosystem is maintained.

**3. Data Sources and Characteristics of SSI**

In accordance with the objectives of the research undertaken, the data source used was represented by the Sustainable Society Index and covered the period 2000–2018, a period for which we have complete data. Starting from these, the indicators used, as well as their values and characteristics, are presented below.

The Sustainable Social Development Index (SSI) includes 21 indicators, integrated into seven categories and aggregated into three dimensions: Human Well-being, Environmental Well-being and Economic Well-being.

The Human Well-being dimension includes three categories of indicators: Basic Needs, Personal Development and Health, and Well-balanced Society, each of them including another three indicators.

The Basic Needs category includes the indicators:

- Sufficient Food—evaluates the quality and sufficiency of feeding; takes values between 2.5 and 10, where the maximum value means that for at least 97.5% of the population, food consumption ensures the level necessary for a normal, active and healthy life.
- Sufficient Drinking Water—evaluates the level of use of basic drinking water services and takes values between 1 and 10, where the maximum value indicates that 100% of the population benefits from these services.
- Safe Sanitation—evaluates the level of use of basic sanitation services; takes values between 1 and 10, where the maximum value indicates that 100% of the population benefits from the use of these services.

The Personal Development and Health category includes the indicators:

- Education—evaluates the gross enrollment rates in primary and secondary education, regardless of age, as a percentage of the age population corresponding to primary and secondary education; takes values between 1 and 10.
- Healthy Life—evaluates the level of life expectancy at birth; takes values between 1 and 10, where the maximum value corresponds to a life expectancy of at least 80 years.
- Gender Equality—based on the index of the gap between the sexes; takes values between 1 and 10, where the maximum value means full equality between the sexes.

The Well-balanced Society category includes the indicators:

- Income Distribution—evaluates the level of equality of distribution among the citizens of a state; takes values between 1 and 10, where the maximum value signifies a perfectly equal distribution.
- Population Growth—evaluates the annual population growth rate; takes values between 1 and 10, where values greater than 8 mean population decline and values less than 8 mean population growth.
- Good Governance—evaluates the level of good governance as the sum of the values of six indicators calculated by the World Bank.

The second dimension of the sustainable society index refers to Environmental Well-being and includes two categories of indicators: Natural Resources and Climate and Energy.

The Natural Resources category includes three indicators:

- Biodiversity—a composite indicator that evaluates the level of biodiversity, taking values between 1 and 10.
- Renewable Water Resources—evaluates freshwater resources extracted in a year as a percentage of the total renewable water resources; takes values between 1 and 10, where the maximum value indicates that the percentage of water regeneration is at least 90%.
- Consumption—evaluates the necessary ecological footprint that an individual needs for the production of resources and the absorption of waste; takes values between 1 and 10, where the minimum value corresponds to a zero ecological footprint requirement.

The Climate and Energy category includes four indicators:

- Energy Use—evaluates the total consumption of primary energy; takes values from 1 to 10, where the maximum value corresponds to zero energy consumption.
- Energy Savings—evaluates the (non-)existence of the reduction of energy consumption over a period of five years; takes values between 1 and 10, where the value of the indicator is 5.5 for constant consumption, for higher values consumption increases, and lower values consumption decreases.
- Greenhouse Gases—evaluates station emissions with greenhouse effect; takes values between 1 and 10, where the maximum value corresponds to an emission level equal to zero.
- Renewable Energy—evaluates the share of energy from renewable sources in the total final energy consumption; takes values from 1 to 10.

The third dimension, Economic Well-being, includes two categories of indicators: Transition and Economy.

The Transition category (the transition to a sustainable society) includes two indicators:

- Organic Farming—evaluates the share of agricultural areas fully converted or in the process of being converted to organic agriculture; takes values between 1 and 10.
- Genuine Saving—evaluates adjusted net savings at the national level; takes values between 1 and 10.

The Economy category includes three indicators:

- GDP—evaluates the volume of gross domestic product per inhabitant; takes values from 1 to 10, where the maximum value corresponds to a GDP/capita greater than $70,000.
- Employment—evaluates the share of employees in the total workforce; takes values from 1 to 10, where the maximum value corresponds to zero unemployment.
- Public Debt—evaluates the level of public debt; takes values between 1 and 10, where the maximum value corresponds to a public debt of less than 2.5% of GDP.

The values corresponding to the categories and the three dimensions are determined by aggregating the values of the indicators/dimensions corresponding to them.

## 4. Methodology

Taking into account the objective of the research, namely, the identification of the similarities and disparities between the European states grouped into the four regions (East, North, South and West), from the point of view of Human Well-being, Environmental Well-being and Economic Well-being, as well as Romania's positioning in this European context, the main methods used were multicriteria analyses, both qualitative and quantitative.

For this, starting from $n = 40$ states included in the analysis and $m = 21$ basic indicators of the SSI, the matrix $Y = \left\| y_{ij} \right\|_{i=\overline{1,n}, j=\overline{1,m}}$ was built.

In addition, seven aggregate indicators presented in Table 1 were included in the analysis.

**Table 1.** Identifiers and meanings of aggregated indicators.

| Variable | Significance | Domain |
|----------|-------------|--------|
| BN | Basic Needs | |
| PDH | Personal Developm. and Health | Human Well-being |
| WBS | Well-balanced Society | |
| NR | Natural Resources | |
| CE | Climate and Energy | Environmental Well-being |
| TRS | Transition | |
| ECN | Economy | Economic Well-being |

Starting from the $Y$ matrix, as well as from the seven indicators aggregated in the different stages of the research, the matrix $Z = \left\| z_{ij} \right\|_{i=\overline{1,n}, j=\overline{1,p}}$ was used, where $p$ represents the number of variables included in the respective cluster analyses. Hierarchical cluster methodology was used to generate the clusters, starting from the $Z$ matrix. Within it, for generating the proximity matrix ($W = \left\| w_{ij} \right\|_{i=\overline{1,n}, j=\overline{1,n}}$), square Euclidian distance was used [65]:

$$W = \left\| w_{ij} \right\|_{i=\overline{1,n}, j=\overline{1,n}}, \ w_{ij} = \sqrt{\sum_{i=1}^{n}\left(z_{ik} - z_{ij}\right)^2}, \ j = \overline{1, p}, \ k = \overline{1, p} \ j \neq i, \ k \neq i, \ w_{ii} = 0 \tag{1}$$

Based on matrix (1), for determining the distance between clusters and their generation, Ward's method was used [66]:

$$\Delta(A, B) = \sum_{i \in A \cup B} \left\| x_i - m_{A \cup B} \right\|^2 - \sum_{i \in A} \left\| x_i - m_A \right\|^2 - \sum_{i \in B} \left\| x_i - m_{\mathbf{B}} \right\|^2 - \frac{n_{A \cap B}}{n_{A \cup B}} \left\| m_A - m_B \right\|^2 \tag{2}$$

In (2), $A$ and $B$ are two clusters, $m_i$ is the centroid, $n_i$ is the number of elements from cluster $i$, and $x_i$ is an item.

Testing the statistical significance of the average values determined at the cluster level and implicitly the correctness of the conclusions drawn from them can be done using the ANOVA methodology. This can only be applied if there are no significant differences between the dispersions of the data series. The testing of this hypothesis was performed with Levene's Test, whose null hypothesis is:

$$H_{0\_1} : \sigma_1^2 = \sigma_2^2 = \sigma_3^2 = \ldots = \sigma_r^2 \tag{3}$$

The condition for accepting the null hypothesis $H_{0\_1}$ (3) is:

$$Sig. F > \alpha \quad equivalent \quad to \quad F_{stat} < F_{\alpha, r-1, n-r} \tag{4}$$

In (4), $F_{stat}$ is the test statistic, $Sig.F$ is its probability, $\alpha$ is the significance threshold, $r$ is the number of clusters and $n$ is the number of variables.

The rejection of hypothesis (3) leads to not using the ANOVA methodology, in which case the Robust Tests of Equality of Means (Welch) was used with the null hypothesis:

$$H_{0\_2} : \mu_1 = \mu_2 = \mu_3 = \ldots = \mu_r \tag{5}$$

In (5), $\mu_i$ represents the average value of the variable at the level of cluster $i$. The condition for accepting the hypothesis $H_{0\_2}$ (5) is:

$$Sig. > \alpha \tag{6}$$

If condition (6) is verified, it follows that the average values recorded at the cluster level do not differ significantly (they are not statistically significant), which leads to the conclusion that the clusters obtained are not relevant, requiring another clustering.

To test the statistical hypotheses (3) and (5), a significance threshold of $\alpha = 0.05$ was used, corresponding to a confidence level of 95%.

The data processing and analysis were carried out using SPSS, and the geographical representations were made by the authors based on GeoDa Windows [67] software developed by Luc Anselin [68] and his team from the Center for Spatial Data Science-The University of Chicago.

## 5. Results and Discussion

The values of the Social Sustainability Index determined by TH Köln–University of Applied Sciences both at the global level and the level of the regions defined, on the one hand by the World Bank, and on the other hand by the United Nations Organization, provide an image of the year-overview of their strengths and weaknesses regarding the transition process towards sustainable development.

### 5.1. Europe versus the World Average

From a geographical point of view, the European continent is contained between the Arctic Ocean to the north, the Mediterranean Sea above, the Atlantic Ocean to the west, and bordered to the east by the Ural Mountains, the Caspian Sea and the Black Sea.

Regarding the organization of the databases of the United Nations (UN), administered by the United Nations Statistics Division (UNSD), the statistical data series are structured in four regions: Eastern Europe, Northern Europe, Southern Europe and West Europe [69]. The values of the social sustainability indices (SSI) were determined according to these regions.

In order to highlight the similarities and differences between the four European regions, the first step in the research was to carry out a comparative analysis between the SSI values of the world average and at the level of each region.

The SSI values recorded at the level of the group of Eastern European states, compared to the world average (World), revealed quite significant fluctuating states. This aspect is very well shown by the spatial graphic representation in Figure 1.

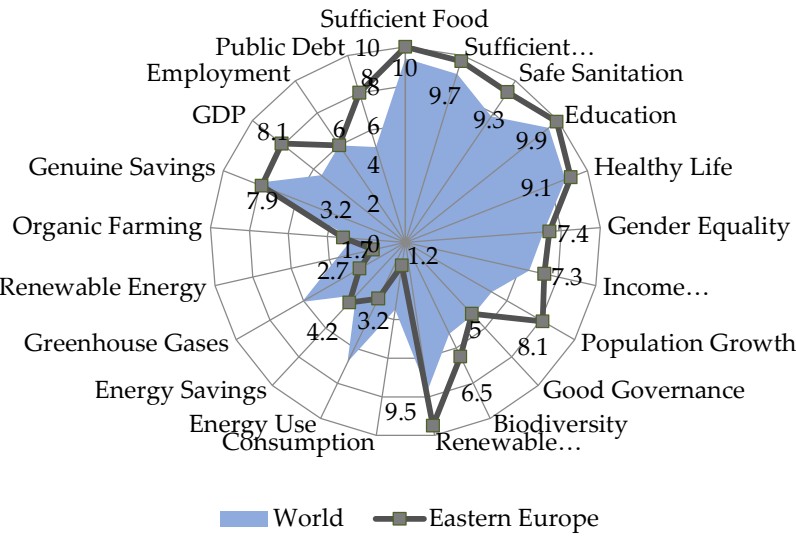

**Figure 1.** Eastern European SSI values versus the world average.

The determined oscillations are highlighted by significant differences between the SSI Values recorded at the level of the group of Eastern European states and the world average (World), which are between −3.6 points for Energy Use and 2.9 points for Public Debt.

In this context, with regard to Human Well-being, one can observe the superiority of the states compared to the world average, established in each category and for each given indicator, which does not always reflect a positive situation from a sustainable point of view. In a favorable situation for sustainability, with positive differences compared to the world average, the indicators are Sufficient Food with 0.6 points, Safe Sanitation with 1.5 points (Basic Needs category), then Education with 0.5 points, Healthy life and Gender Equality with 0.3 points each (Personal Development and Health category) and from the Well-balanced Society category, only Income Distribution with 0.8 points. In this last category, Good Governance faces a reduction compared to the world average by 0.1 points, which highlights the problems that negatively influence sustainability at the level of Eastern Europe. For the Sufficient Drinking Water and Population Growth indicators for Eastern Europe, although they face a value increase compared to the world average by 0.7 and 3 percentage points, the high values of 9.7 and 8.1 indicate unfavorable situations in terms of poor usage of water services and population reduction.

Accelerated population aging, health financing and provision, and prohibitively expensive medical technologies represent the core sustainability challenges in Eastern European and Balkan countries [70]. The accessibility and quality of the healthcare system could be improved through health policies that put patients at the center of all concerns [71]. On the other hand, the practice of preventive medicine is vital for maintaining quality of life as much as possible, with as little costs as possible.

Three indicators of the Environmental Well-being category have values lower than the world average, which reflects unfavorable situations for sustainability. Thus, it can be highlighted that Energy Use is 3.6 points below the world average, Greenhouse Gases by 3.4 points and Renewable Energy by 1.9 points. The value reduced by 2.3 points compared to the world average of the Consumption indicator translates to a favorable situation for Eastern Europe, as well as the 1.2 points being very close to 1, which represents the required zero ecological footprint. At the same time, in this category, there are also positive differences compared to the average, which indicates positions favorable for sustainability. Thus, in addition to the difference of 1.3 points for Biodiversity and 1.8 points for Renewable Water Resources, the difference from the average of 0.5 points for Energy Savings also indicates a favorable sustainable situation through the value of 4.2 points less than the 5.5 established and presented theoretically.

Two indicators from the Transition category register values at the level of Eastern Europe above those of the world average, showing their efficiency from a sustainable point of view. Thus, GDP is placed above the average by 2.6 points and Organic Farming by 0.5 points. Eastern Europe's employment is six points, which is the same as the world average. The Genuine Savings indicator, with a value of 7.9 points, is 0.4 points below the average, although it is in a favorable situation for sustainability. There is an exception to the Public Debt indicator because although the difference from the average is 2.9 points, it does not indicate a positive state of sustainability at the level of Eastern Europe, through the high value recorded, of 8 points.

Similarly, as for Eastern Europe, a spatial figure was created for Northern Europe as well (Figure 2), which more clearly shows the similarities and differences between the SSI values of the region and the world average.

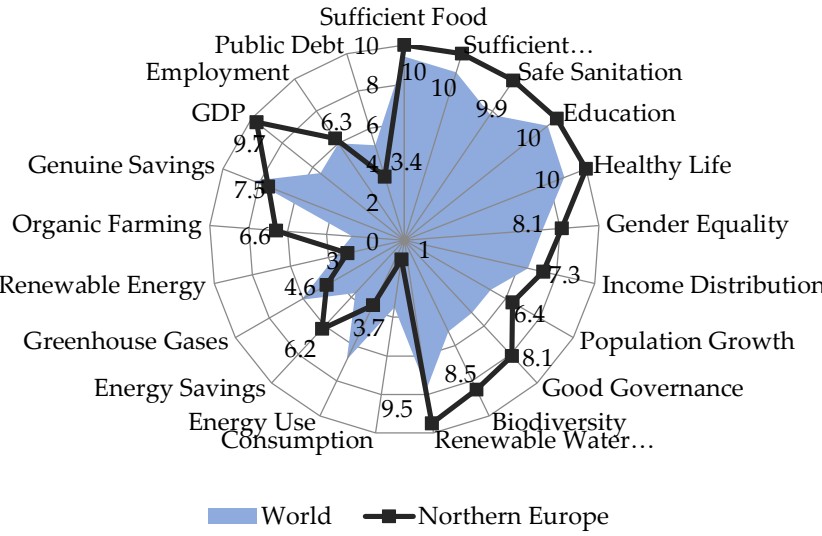

**Figure 2.** Northern European SSI values versus the world average.

In the case of the group of Northern European countries (Figure 2), the most significant differences compared to the World average are recorded in the GDP indicators with 4.2 points, Organic Farming with 3.9 points, Biodiversity with 3.3 points, and Good Governance with 3 points. Through the values recorded by these indicators, the positive situation in the manifestation of sustainability at the level of this European region is highlighted. Indicators also in favorable positions for sustainability are the indicators with a percentage difference between 1 and 3 points: Safe Sanitation with 2.1 points, Renewable Water Resources with 1.8 points, Population Growth with 1.3 (the value of 6.4 points being lower than the theoretical value of 8 points), Healthy Life with 1.2 points, Gender Equality with 1 point and Sufficient Drinking Water with 1 point, for which 10 points indicate that 100% of the Northern European population benefits from these services.

Subunit values of the differences between the values recorded at the level of the group of Eastern European states (Figure 1) compared to the world average are established for Income Distribution (0.8 points), Education and Sufficient Food (with 0.6 points), as well as for Employment with 0.3 points, and these indicators show values which highlight the positive effect from the point of view of sustainability. It can also be mentioned that the maximum value of 10 recorded for Sufficient Food indicates that for at least 97.5% of the population of Northern Europe, food consumption provides the necessary level for a normal, active and healthy life, which highlights the good situation of this indicator in the development of the sustainability process. Although the difference from the average is 2.5 points, the Energy Savings, with a value of 6.2 points higher than the theoretical value of 5.5 points, indicates an increased energy consumption, therefore, an unfavorable situation at the level of Northern Europe in terms of Environmental Well-being. Both through the recorded values and the reported differences, which are below the average value, three indicators show the unfavorable state in the direction of sustainability. Thus, for Energy Use, the negative difference of 3.1 points and the value of 3.7 points indicate that at the level of Northern Europe, there is high primary energy consumption. Additionally, for Greenhouse Gases, the 4.6 points indicate a rather high level of emissions for Northern Europe, the negative difference of 1.5 points confirms this unfavorable situation for sustainability. The high consumption of renewable energy in Northern Europe is confirmed by the low percentage transposed into the 3 points, 0.6 points less than the 3.6 set world average. From the Natural Resources group, a component of Environmental Well-being, with a Consumption value of 1 point recorded, alt-

hough it is lower by 2.5 points compared to the world average, it is, at the level of Northern Europe, in the best situation for sustainability because this minimum value corresponds to a zero ecological footprint requirement. Northern Europe is also in a positive state with regard to the Real Economies indicator, which, even if the score of 7.5 is 0.8 points below the world average, the value still confirms the positive sustainable situation. The Public Debt, by 3.4 points, is below the world average, which leads to a difference of 1.7 points, reflecting a level of public debt higher than 2.5% of the GDP, thus rated as unfavorable for sustainability.

Compared to the world average of the SSI, the values of this indicator at the level of the third region (Southern Europe) register significant fluctuations, a fact more clearly highlighted by the spatial graphic representation (Figure 3). Thus, the differences between the SSI values recorded at the level of the group of Southern European states and the world average (World) are between −3.4 points for Public Debt and 5.7 points for Organic Farming.

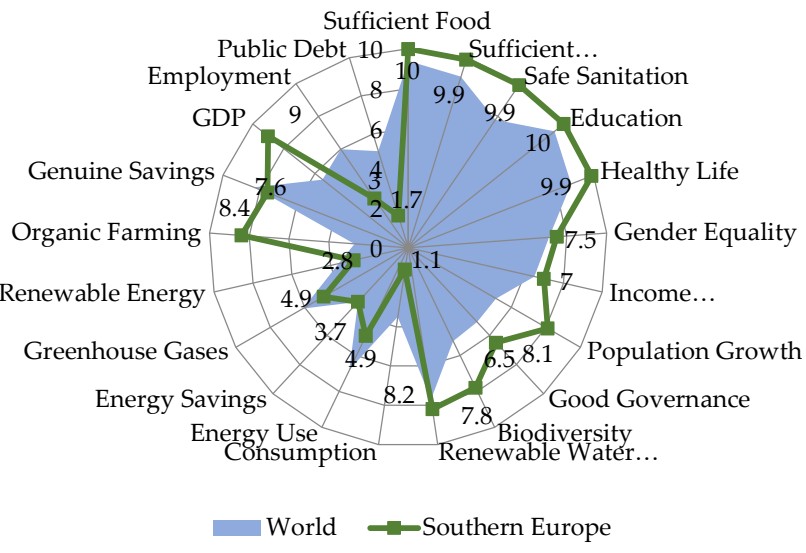

**Figure 3.** Southern Europe SSI values versus the world average.

The negative differences between the SSI values registered at the level of the group of Southern European states and the world average are recorded for Employment at −3 points, for Consumption at −2.4 points, for Energy Use at −1.9 points and for Greenhouse Gases at −1.2 points. The negative subunit differences of −0.8 points (Renewable Energy) and −0.7 points (Genuine Savings) are added and the positive subunit differences observed for Gender Equality (0.4 points), Income Distribution and Renewable Water Resources (0.5 points), Sufficient Food (0.6 points), and Sufficient Drinking Water (0.9 points) are highlighted.

Next, significant differences are determined for Healthy Life (1.1 points), Good Governance (1.4 points), Safe Sanitation (2.1 points), Biodiversity (2.6 points), Population Growth (3 points), which, by the high value of 8.1, indicates an unfavorable situation in terms of population reduction, as well as GDP by 3 points. For Energy Savings, a value of 3.7 is equal to that of the world average, indicating a reduced consumption, therefore, a favorable situation.

Regarding the ratios between the SSI values recorded at the level of the fourth group (Western Europe) and the world average, there also are large differences, as shown in the spatial graphic representation (Figure 4).

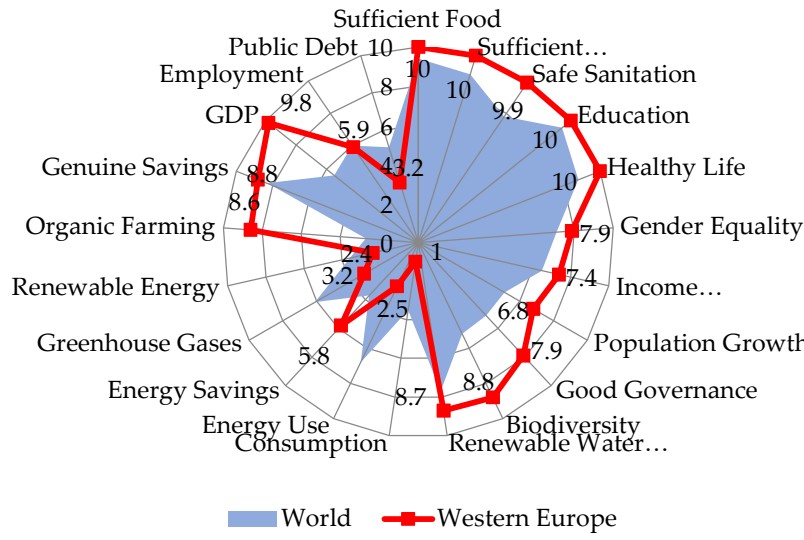

**Figure 4.** Western European SSI values versus the world average.

The biggest differences range between 1 point recorded for Renewable Water Resources, respectively, Sufficient Drinking Water, and 5.9 points for Organic Farming, as follows: 4.3 points for GDP, then 3.6 points for Biodiversity, followed by 2.8 points for Good Governance, the same 2.1 points for Safe Sanitation and Energy Savings, then 1.7 points for Population Growth (the high value of 6.8 means a favorable population growth situation) and only 1.2 points for Healthy Living. Small, positive subunit differences belong to Income Distribution (0.9 points), Gender Equality (0.8 points), Sufficient Food and Education (0.6 points) and Genuine Savings (0.5 points).

Between the values recorded at the level of the group of states in Western Europe and the world average, there were also negative values of the differences between −0.1 points belonging to Employment and −4.3 points for Energy Use, as follows: for Renewable Energy a difference of −1.2 points was determined, for Public Debt −1.9 points, Consumption −2.5 points and Greenhouse Gases −2.9 points.

*5.2. Similarities and Disparities between the European States*

From the research and analysis carried out between the four European regions, there are both similarities and differences regarding the values and implicitly the stage of development of sustainable societies in the European continent, a fact noted by the results obtained as a result of a more detailed analysis by clustering, on the three aggregate indicators of the SSI (Human Well-being, Environmental Well-being, Economic Well-being), representing the second step in the study carried out.

The first area analyzed is Human Well-Being, and the overview of SSI in Europe is shown in Figure 5.

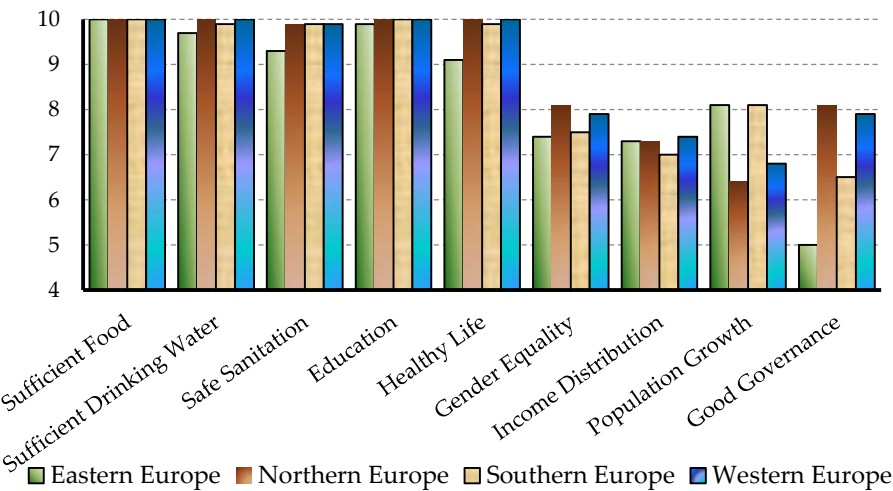

**Figure 5.** Human Well-being in Europe considering SSI indicators.

The Human Well-being dimension (Figure 5) can be characterized by similarities between the four regions (Eastern Europe, Northern Europe, Southern Europe and Western Europe) in terms of Sufficient Food, the maximum score highlighting that, at least for 97.5% of the population of each region, food consumption ensures the level necessary for a normal, active and healthy life, a situation favorable for sustainability. Similarities between three regions (Northern Europe, Southern Europe and Western Europe) are observed for Education, with a maximum score of 10 (Eastern Europe receiving 9.9 points), as well as for Safe Sanitation with 9.9 points (9.3 points recorded for Eastern Europe). With a maximum score of 10, similarities are observed between two regions (North and West) in Sufficient Drinking Water and Healthy Life. Each indicator registers 9.9 points at the level of Southern Europe, while for Eastern Europe, 9.7 points are reported for Sufficient Drinking Water and 9.1 for the second indicator.

Low scores, with values between 5 and 8.1 points, are recorded for the other indicators in all four regions. The most significant difference between regions (3.1 points) is reported for the Good Governance indicator, with a minimum of 5 points recorded by Eastern Europe (characterized by a low level of Good Governance) and a maximum of 8.1 points in Northern Europe. The high level of Good Governance positively stimulates sustainability. Another quite significant difference, of 1.7 points, belongs to the population growth indicator with scores of 6.4 (Northern Europe) and 6.8 (Western Europe), indicating an annual Population Growth rate favorable to sustainability, while population reduction is given by the score of 8.1 corresponding to Eastern and Southern Europe. Gender equality is another indicator with different scores from one region to another, with a difference of 0.7 points obtained between the maximum of 8.1 in Northern Europe and 7.4 in Eastern Europe. With a difference of only 0.4 points, the Income Distribution indicator stands out, for which the points by region are reduced, oscillating between 7 points for Eastern Europe and 7.3 points for Eastern and Northern Europe, respectively, and 7.4 points for Western Europe.

The cluster analysis of the values of the synthetic indicators of the Human Well-being dimension (Basic Needs, Personal Developm. and Health, and Well-balanced Society) highlights a structure of six clusters (Table 2) with significant differences.

**Table 2.** Structure of clusters determined by Basic Needs, Personal Developm. and Health, and Well-balanced Society synthetic indicators of the Human Well-being Dimension.

| Cluster | Countries | Region |
|---|---|---|
| C1 | Czech Republic, Hungary, Poland, Romania, Slovak Republic | Eastern Europe |
| | Greece, Italy, Portugal, Serbia, Slovenia, Spain | Southern Europe |
| | Denmark, Estonia, Finland, Ireland, Latvia, Lithuania, Norway, Sweden, United Kingdom | Northern Europe |
| | Belgium, France, Germany, The Netherlands, Switzerland | Western Europe |
| C2 | Belarus, Russian Federation | Eastern Europe |
| | Albania, Cyprus | Southern Europe |
| | Luxembourg | Western Europe |
| C3 | Bulgaria, Moldova, Ukraine | Eastern Europe |
| | Croatia, North Macedonia | Southern Europe |
| C4 | Iceland | Northern Europe |
| C5 | Bosnia and Herzegovina | Southern Europe |
| C6 | Malta, Montenegro | Southern Europe |

A more detailed and clearer picture of the spatial distribution, from the point of view of Human Well-being between the formed clusters and, therefore, implicitly between the countries and regions of Europe, is highlighted more clearly in Figure 6.

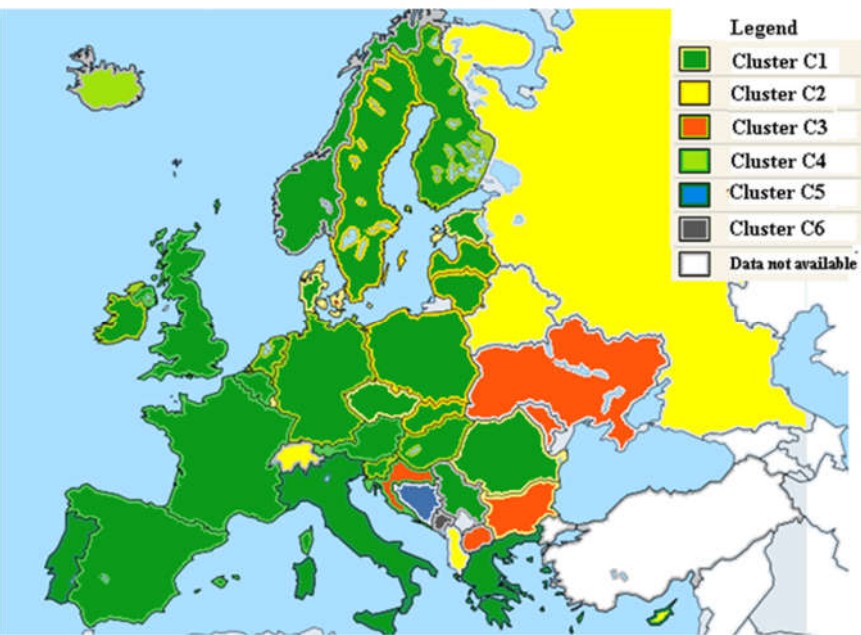

**Figure 6.** Spatial distribution of clusters, from the point of view of the Human Well-being Dimension.

Testing the membership of the countries, implicitly of the regions to the formed clusters, requires testing the non-existence of significant differences between the disparities of the series by applying Levene's Test.

Following the application of Levene's Test, the results (Table 3) through sig. values lower than 0.05 (0.00 and 0.03) show us that there are significant differences between the dispersions determined by clusters, so ANOVA cannot be applied. In this context, the Welch robustness test was used.

**Table 3.** Test of Homogeneity of Variances of the variables of the Human Well-being Dimension.

|  | Levene Statistic | df1 | df2 | Sig. |
|---|---|---|---|---|
| BN | 68.171 | 3 | 34 | 0.000 |
| PDH | 13.598 | 3 | 34 | 0.000 |
| WBS | 5.500 | 3 | 34 | 0.003 |

Studying the results of the Welch robustness test in Table 4, one can note the significant difference between the averages of the variables at the level of each cluster, a fact confirmed by the values of sig. < 0.05, highlighting the belonging of each variable (BN, PDH, WBS) to the determined clusters.

**Table 4.** Robust Tests of Equality of Means of the variables of the Human Well-being Dimension.

|  |  | Statistic [a] | df1 | df2 | Sig. |
|---|---|---|---|---|---|
| BN | Welch | 11.092 | 3 | 3.494 | 0.028 |
| PDH | Welch | 45.009 | 3 | 3.784 | 0.002 |
| WBS | Welch | 58.630 | 3 | 3.438 | 0.002 |

[a]. Asymptotically F distributed.

The presentation of the similarities and disparities of this dimension of Human Well-being is continued by the analysis of the characteristics of the clusters noted by the average values determined on the variables and clusters (Table 5).

**Table 5.** Characteristics of clusters of the Human Well-being Dimension.

| Variable | C1 | C2 | C3 | C4 | C5 | C6 |
|---|---|---|---|---|---|---|
| BN | 9.9192 | 9.8200 | 8.6400 | 5.5000 | 3.4000 | 7.8500 |
| PDH | 9.1000 | 8.9000 | 7.0000 | 8.2000 | 3.6000 | 2.9500 |
| WBS | 7.3692 | 6.2600 | 4.2800 | 3.4000 | 1.1000 | 2.9000 |

The average values obtained highlight that the most significant disparities are those regarding Personal Developm. and Health (Figure 7) and Well-balanced Society (Figure 8).

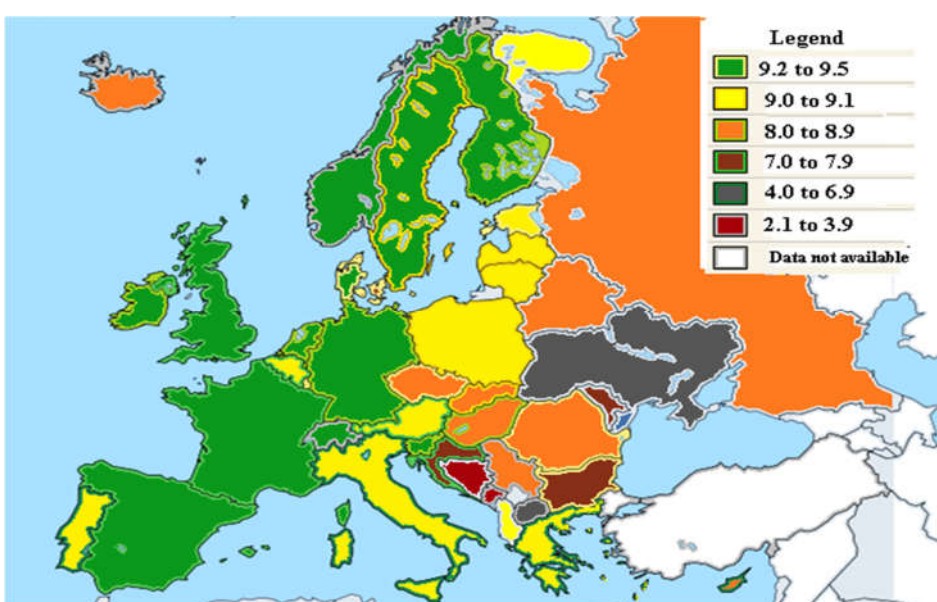

**Figure 7.** Grouping of European states according to Personal Developm. and Health.

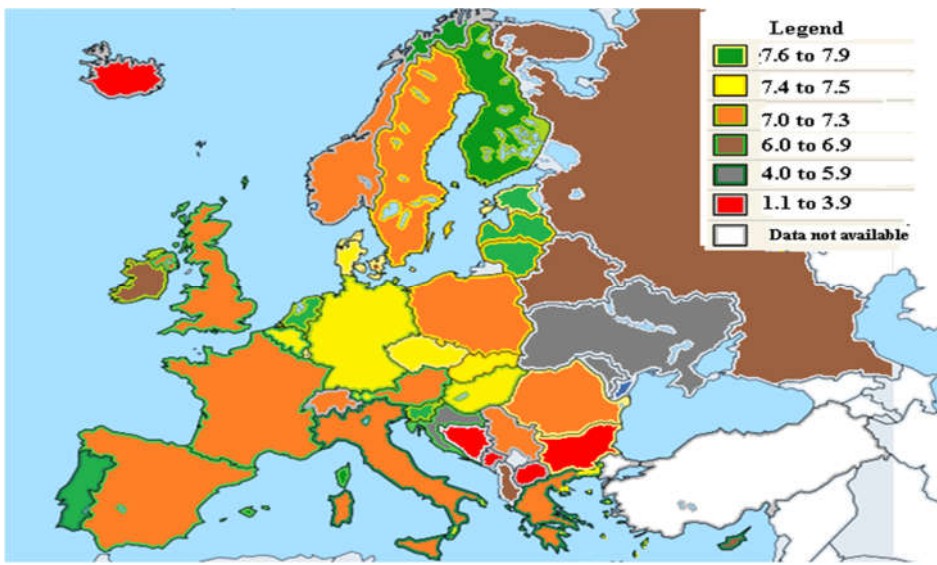

**Figure 8.** Grouping of European states from the point of view of Well-balanced Society.

The significant disparities can be explained and observed as a result of the oscillations which appear both at the level of the six formed groups as well as between the clusters, a fact highlighted by the length of the variation in the interval of the Personal Developm. and Health variable of 6.5 average points. Cluster 6, formed only by Malta and Montenegro belonging to the Southern Europe region, corresponds to the lowest average value of 2.95 points of the analyzed variable. It should be noted that, in Cluster 1, most countries from all regions of Europe are included (Czech Republic, Hungary, Poland, Romania, and Slovak Republic from Eastern Europe; Greece, Italy, Portugal, Serbia, Slovenia, and Spain from Southern Europe; Denmark, Estonia, Finland, Ireland, Latvia, Lithuania, Norway, Sweden, and United Kingdom from Northern Europe; Belgium, France, Germany, The Netherlands, and Switzerland from Western Europe) and the maximum average score is recorded both for the variable Personal Developm. and Health, as well as for Well-balanced Society. Bosnia and Herzegovina, in Southern Europe (cluster 5), registers the minimum average value of 1.1 points for Well-balanced Society.

The analysis of the stage of development of sustainable societies in the European continent is completed by the interpretation of the dimension of Environmental Well-being in Europe considering the SSI indicators (Figure 9).

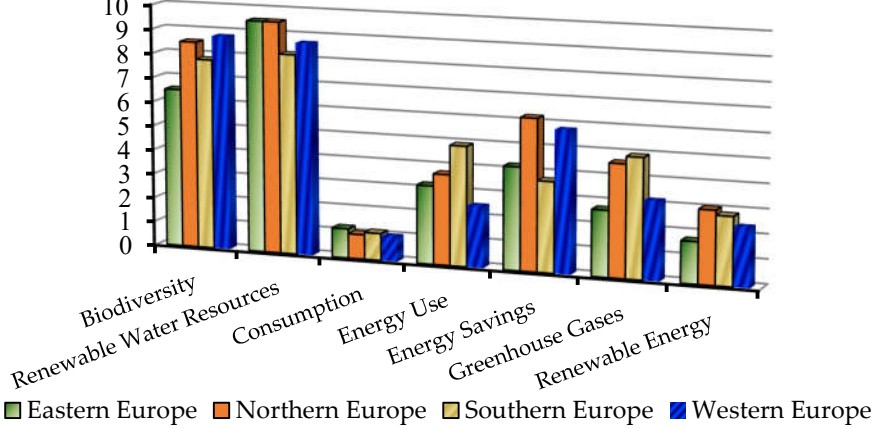

**Figure 9.** Environmental Well-being in Europe considering SSI indicators.

Regarding Natural Resources, as the first component of Environmental Well-being, it can be mentioned that all three indicators have scores that highlight a favorable impact on sustainability. Between the four European regions (Eastern Europe, Northern Europe, Southern Europe and Western Europe), the biggest gap appears in Biodiversity (2.3 points), with a maximum of 8.8 points recorded in Western Europe and a minimum of 6.5 in Eastern Europe. Renewable Water Resources is the indicator with a difference of 1.3 points between regions, with the highest score of 9.5 belonging to Eastern Europe and the lowest of 8.2 points to Southern Europe. Although the lowest score between 1 (Northern Europe, Western Europe) and 1.2 (Eastern Europe) with the smallest difference (0.2 points) belongs to the Consumption indicator, these values nevertheless indicate a situation favorable for the sustainability process (where a score of 1 indicates the necessary zero ecological footprint).

The second component of Environmental Well-being, Climate and Energy, includes four indicators (Energy Use, Energy Savings, Greenhouse Gases and Renewable Energy), which are analyzed by regions in relation to their scores. Thus, at the level of the four regions, the Energy use indicator has low scores that oscillate between 2.5 points (Western Europe) and 4.9 points (Southern Europe) which are not favorable for the sustainability process, with 1 representing the highest total consumption of primary energy. In Energy Savings, Eastern Europe, with 4.2 points, and Southern Europe, with 3.7 points, succeed in terms of sustainability by this indicator, a fact noted by the reduced values below 5.5 points. Western Europe, with 5.8 points, and Northern Europe, with 6.2 points, are not in a favorable situation in terms of sustainability. The low scores recorded at the European level for the Gas Emissions indicator are between 2.7 points for Eastern Europe and 4.9 points for Southern Europe. They signal problems at the European level regarding the manifestation of the sustainability process considering that the minimum score of indicating the highest level of gas emissions. In addition, with regard to the Renewable Energy indicator, the situation is similar to that presented in the previous indicator (Greenhouse Gases). Thus, we can specify the small fluctuating scores between 1.7 points in Eastern Europe and 3 points in Northern Europe, signaling the negative impact on sustainability, regardless of the region.

Detailing the study by analyzing the values of the synthetic indicators of the Environmental Well-being dimension (Natural Resources, Climate and Energy) by clusters and countries, the results also indicate a six clusters structure (Table 6).

**Table 6.** Structure of clusters determined by Natural Resources and Climate and Energy systematic indicators of the Environmental Well-being Dimension.

| Cluster | Countries | Region |
|---------|-----------|--------|
| C1 | Czech Republic, Hungary, Poland, Russian Federation | Eastern Europe |
| | Cyprus, Serbia | Southern Europe |
| | Estonia, Finland, Ireland | Northern Europe |
| | Belgium, Germany, Luxembourg, The Netherlands | Western Europe |
| C2 | Slovak Republic | Eastern Europe |
| | Lithuania, Norway, Sweden, United Kingdom | Northern Europe |
| | Greece, Italy, Montenegro, Spain | Southern Europe |
| | Austria, France | Western Europe |
| C3 | Belarus, Romania | Eastern Europe |
| | Denmark, Latvia | Northern Europe |
| | Portugal, Slovenia | Southern Europe |
| | Switzerland | Western Europe |
| C4 | Bulgaria | Eastern Europe |
| | Iceland | Northern Europe |
| C5 | Moldova, Ukraine | Eastern Europe |
| | Albania, North Macedonia | Southern Europe |
| C6 | Bosnia and Herzegovina, Croatia | Southern Europe |

From the point of view of Environmental Well-being between the formed clusters respectively between the countries and regions of Europe, the spatial distributions are shown more clearly in Figure 10.

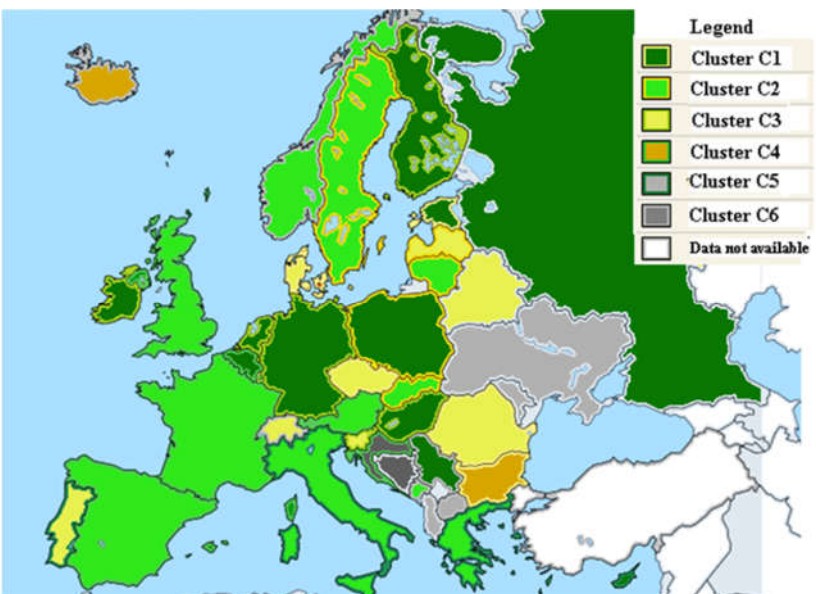

**Figure 10.** Spatial distribution of clusters from the point of view of Environmental Well-being.

According to the presented methodology, the analysis continues with testing the membership of the countries, implicitly of the regions to the clusters.

The application of Levene's Test led to the obtaining of some results (Table 7), which, through the values of sig. lower than 0.05 (0.00 and 0.098) show us that there are significant differences between the dispersions determined on the clusters. Since ANOVA cannot be applied in this context, the Welch robustness test was used.

**Table 7.** Test of Homogeneity of Variances of the variables of the Environmental Well-being Dimension.

|    | Levene Statistic | df1 | df2 | Sig. |
|----|----|----|----|----|
| NR | 14.014 | 5 | 33 | 0.000 |
| CE | 2.041 | 5 | 33 | 0.098 |

The results of applying the Welch robustness test (Table 8) indicate a significant difference between the means of the Natural Resources and Climate and Energy variables at the level of each cluster. This statement is confirmed by the sig. values of 0.044 and 0.000, which are less than 0.05, which highlights the membership of each mentioned variable to the determined clusters.

**Table 8.** Robust Tests of Equality of Means of the variables of the Environmental Well-being Dimension.

|    |    | Statistic [a] | df1 | df2 | Sig. |
|----|----|----|----|----|----|
| NR | Welch | 6.891 | 5 | 4.983 | 0.044 |
| CE | Welch | 751.294 | 5 | 8.117 | 0.000 |

[a]. Asymptotically F distributed.

The similarities and disparities of the Environmental Well-being Dimension are also highlighted by presenting the average values determined by variables and clusters (Table 9), highlighting their characteristics.

**Table 9.** Cluster characteristics of the Environmental Well-being Dimension.

| Variable | C1 | C2 | C3 | C4 | C5 | C6 |
| --- | --- | --- | --- | --- | --- | --- |
| NR | 4.1231 | 4.2182 | 4.2000 | 4.9000 | 5.5250 | 3.9500 |
| CE | 2.2769 | 3.7091 | 4.6143 | 6.7250 | 4.0500 | 8.5500 |

The Natural Resources variable registers a rather small oscillation between countries and regions of 1.58 points, with the maximum registered by cluster 5 (Moldova, Ukraine from Eastern Europe and Albania, North Macedonia from Southern Europe) and the minimum by cluster 6 (Bosnia and Herzegovina, Croatia from Southern Europe). If cluster 6 has the lowest average score for Natural Resources, it will record the highest for the other Environmental Well-being variable, Climate and Energy. The situation of this Climate and Energy variable presents itself differently considering the amplitude of 6.27 points determined based on the maximum recorded by cluster 1 represented by most countries, Czech Republic, Hungary, Poland, and Russian Federation from Eastern Europe; Cyprus, Serbia from Southern Europe; Estonia, Finland, and Ireland from Northern Europe; and Belgium, Germany, Luxembourg, and The Netherlands from Western Europe.

Another addition to the analysis of the stage of development of sustainable societies in the European continent is achieved by rendering the image of Economic Well-being in Europe considering SSI indicators (Figure 11).

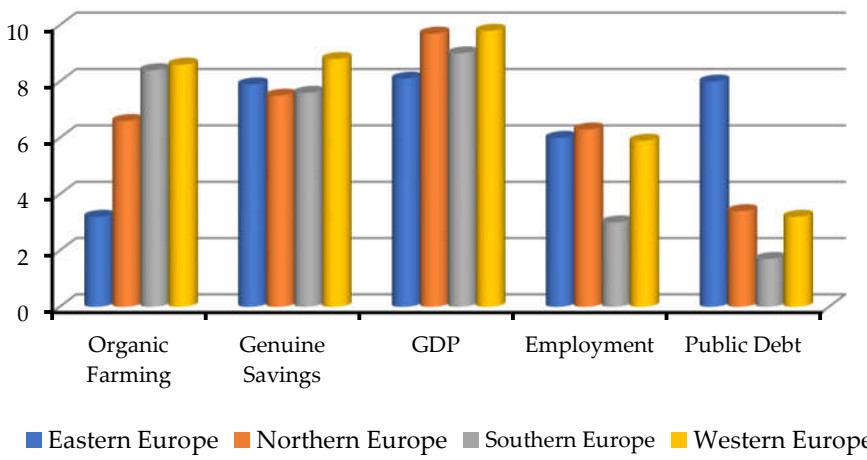

**Figure 11.** Economic Well-being in Europe considering SSI indicators.

Among the five SSI indicators that are components of Economic Well-being, Organic Farming is the indicator facing the biggest gap at the European level (5.4 points). Northern Europe, Southern Europe and Western Europe, with scores from 6.6 to 8.6 points, represent the regions that, through this indicator, have a positive impact on sustainability. At the opposite pole is Eastern Europe in the sense that, with 3.2 points, it is in an unfavorable situation in the development of the sustainability process. The high scores between 7.5 points for Northern Europe and 8.8 points for Western Europe cause a gap of 1.3 points for the Genuine Savings indicator, being in a favorable position for sustainable development. A similarly positive situation for sustainable development is reported at the level of all four regions for the Economy indicator, transposed through GDP. Thus, the differ-

ence between regions of 1.7 points is the result of the oscillations recorded between Western Europe, which corresponds to 9.8 points, and Eastern Europe, with 8.1 points. Regarding Employment, the situation is a little different in the sense that it is favorable for sustainable development only for Western Europe (5.9 points), Eastern Europe (6 points) and Northern Europe (6.3 points). Southern Europe is in an unfavorable situation for sustainable development, a fact confirmed by the low score of 3 points registered for Employment and the 1.7 points (value less than 2.5 points) for the Public Debt indicator. Therefore, the same position favorable for sustainability is signaled for the other three regions regarding the Public Debt indicator, the values oscillating between 3.2 points (Western Europe) and 8 points (Eastern Europe), confirming the stated affirmation.

A structure also of six clusters (Table 10) is obtained by applying the clustering methodology to the Economic Well-being Dimension represented by two components: Transition and Economy.

**Table 10.** Structure of clusters determined by Transition and Economy synthetic indicators of the Economic Well-being Dimension.

| Cluster | Countries | Region |
|---------|-----------|--------|
| C1 | Czech Republic, Slovak Republic | Eastern Europe |
| | Denmark, Estonia, Latvia, Lithuania, Sweden | Northern Europe |
| | Germany, Switzerland | Western Europe |
| C2 | Hungary | Eastern Europe |
| | Finland, Iceland, Ireland | Northern Europe |
| | Slovenia | Southern Europe |
| | Austria | Western Europe |
| C3 | Cyprus, Italy, Montenegro, Portugal, Spain | Southern Europe |
| | Belgium, France | Western Europe |
| C4 | Bulgaria, Poland, Moldova | Eastern Europe |
| | Norway | Northern Europe |
| | Malta | Southern Europe |
| | Luxembourg, The Netherlands | Western Europe |
| C5 | Belarus, Romania, Russian Federation, Ukraine | Eastern Europe |
| | Croatia, North Macedonia | Southern Europe |
| C6 | United Kingdom | Northern Europe |
| | Albania, Bosnia and Herzegovina, Greece, Serbia | Southern Europe |

The spatial distribution of the clusters from the point of view of Economic Well-being clearly highlights the differences between the countries and regions of Europe through the appropriate graphical representation (Figure 12).

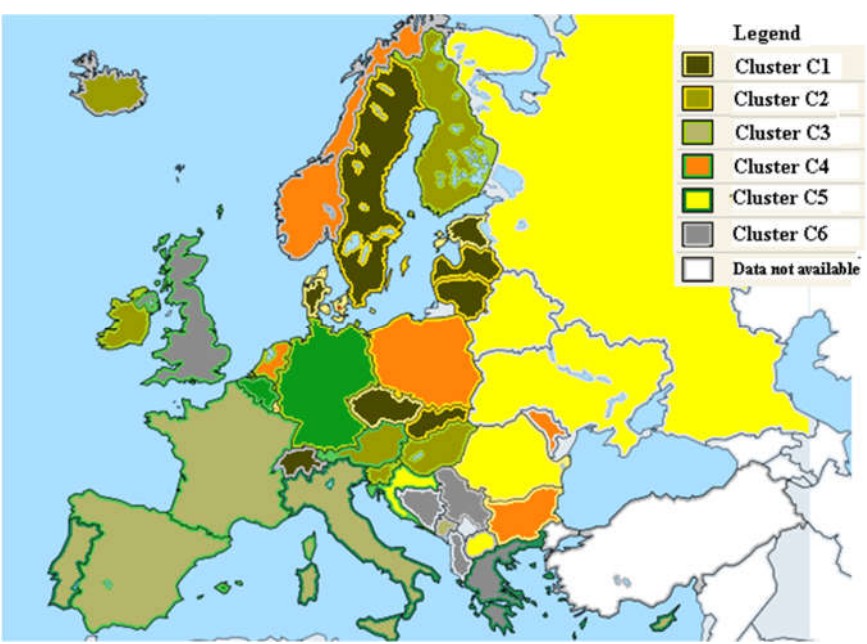

**Figure 12.** Spatial distribution of clusters from the point of view of Economic Well-being.

Testing the homoscedasticity of the dispersions of the data series to highlight the belonging of the countries, implicitly the regions, to the clusters was carried out by applying Levene's Test.

Considering the results obtained by applying Levene's Test (Table 11), and therefore the insignificance of the results in Table 3 (the sig. values are lower than the significance level of 0.05), all the dispersions within the groups (Transition and Economy) are not homogeneous. In this context, the Welch robustness test still applies.

**Table 11.** Test of Homogeneity of Variances of the variables of the Economic Well-being Dimension.

|  | **Levene Statistic** | **df1** | **df2** | **Sig.** |
|---|---|---|---|---|
| TRS | 3.014 | 5 | 34 | 0.023 |
| ECN | 1.058 | 5 | 34 | 0.040 |

The values of the Welch test statistic (Table 12) for both variables (Transition and Economy) are significant considering that the sig. values are less than 0.05, thus, indicating that the averages of the variables differ significantly at the level of each cluster. Following the application of the methodology, the results highlighted the membership of the Transition variable and the Economy variable to the determined clusters.

**Table 12.** Robust Tests of Equality of Means of the variables of the Economic Well-being Dimension.

|  |  | **Statistic [a]** | **df1** | **df2** | **Sig.** |
|---|---|---|---|---|---|
| TRS | Welch | 41.547 | 5 | 13.725 | 0.000 |
| ECN | Welch | 103.729 | 5 | 14.560 | 0.000 |

[a]. Asymptotically F distributed.

The study of the dimension of Economic Well-being is further reproduced by presenting the similarities and disparities of the characteristics of the clusters, taking into account the average values obtained for each variable and cluster (Table 13).

**Table 13.** Characteristics of clusters of the Economic Well-being Dimension.

| Variable | C1 | C2 | C3 | C4 | C5 | C6 |
|---|---|---|---|---|---|---|
| TRS | 9.0778 | 8.4167 | 8.3000 | 7.5000 | 4.2167 | 4.0600 |
| ECN | 7.6000 | 5.4333 | 3.3000 | 7.8000 | 7.9333 | 3.6000 |

In the case of the Transition component of Economic Well-being, the scores oscillate between an average of 4.06 for the component countries of cluster 6 (United Kingdom from Northern Europe and Albania, Bosnia and Herzegovina, Greece, Serbia from Southern Europe) and 9.08 for the countries which constitute cluster 1 (Czech Republic, Slovak Republic from Eastern Europe; Denmark, Estonia, Latvia, Lithuania, Sweden from Northern Europe; and Germany, Switzerland from Western Europe) with an amplitude of 5.02 points. For the other component of Economic Well-being, Economy, the length of the variation range is 4.5 points less compared to that of the previous component, where the scores oscillate between the Cluster 3 average (Cyprus, Italy, Montenegro, Portugal, Spain from Southern Europe and Belgium and France from Western Europe) and that of cluster 4 (Bulgaria, Poland, Moldova from Eastern Europe; Norway from Northern Europe; Malta from Southern Europe; and Luxembourg, The Netherlands from Western Europe).

The disparities in the level of sustainability affect the existence of an efficient economic system. In order to stimulate inclusion at the European level, a common strategy is needed to achieve intelligent and sustainable integrated growth [72].

*5.3. Social Sustainability in Romania in the Context of the European Union*

From the point of view of the values of the social development index, among the member states of the European Union, Romania is characterized by a wide range of values that signify both strong points and weak points in sustainable economic and social development.

5.3.1. An Overview of Romania's Place among the Member States of the European Union

An overview of Romania's place among the member states of the European Union considering SSI values is provided in Figure 13.

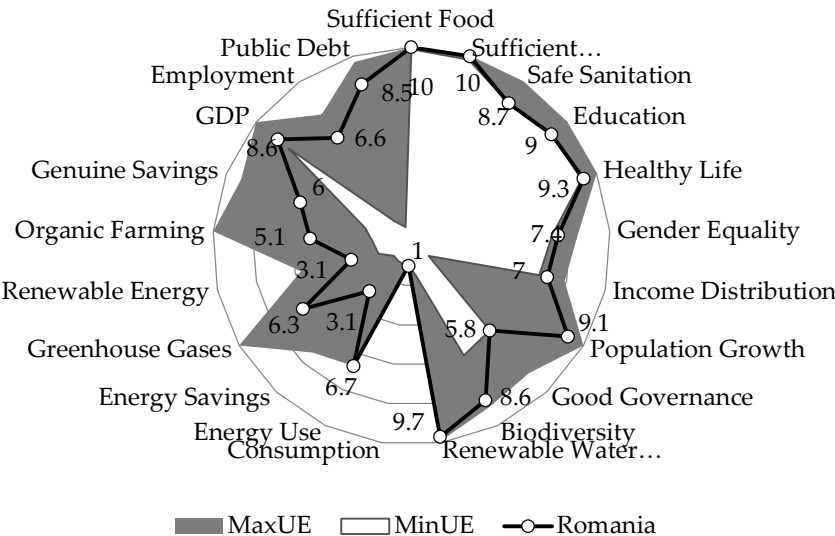

**Figure 13.** Romania's place in the range of SSI values registered in the EU27 states.

Romania, through the 10 points obtained for the Sufficient Food and Sufficient Drinking Waters indicator, presents itself in a favorable situation from the point of view of sustainable development, representing the Max EU. The situation is not as favorable for Romania in regard to the Safe Sanitation indicator with a value of 8.7, as it represents the Min EU, and compared to the Max EU of 10 points, there is a difference of 1.3 points.

With regard to the Education, Healthy Life and Gender Equality indicators, it can be stated that Romania presents itself in a satisfactory position compared to MaxUe and Min EU, despite the fact that through the recorded values, each individual indicator has a positive impact on sustainable development. Thus, through the 9 points recorded for Education, Romania is at the Min EU value and 1 point difference from the Max EU value. The score of 9.3 for Healthy Life highlights that Romania is 0.7 points below MaxEU and 0.1 above MinEU. The score of the Gender Equality indicator (7.4 points) for Romania is 1 point less than MaxEU and 0.3 more than MinEU.

In Romania, quite satisfactory results are recorded, compared to EU Max and EU Min, by an Income Distribution score of 7 points (score lower by 0.9 compared to MaxEU and higher by 0.4 compared to MinEU) and Population Growth of 9.1 points (also lower by 0.9 compared to MaxUE and higher by 1 compared to Min EU). The Good Governance indicator is not in the same favorable position, as 5.8 points are the level of the Min EU, although, for sustainability, it represents a good influence.

Regarding the Environmental Well-being indicators, it can be mentioned that all three components of Natural Resources, with their values of 8.6 points (Biodiversity), 9.7 points (Renewable Water Resources) and 1 point (Consumption), approach Max EU and even achieves it for Consumption, which highlights the quite positive impact on the development of the sustainable society.

Although the differences from Max EU are more significant compared to the other category, all the Climate and Energy indicators are on the same positive trajectory in terms of sustainability. Thus, if an Energy Use score of 6.7 is identical to MaxUE, and in Greenhouse Gases and Renewable Energy, the differences of 3.7 points and 2.6 points compared to the MaxEU are acceptable for positive influences on sustainability. We cannot mention much about the difference of 4.2 points determined in Energy Savings, but we can highlight that the much lower value of this indicator at the level of Romania compared to MaxEU is significant because it indicates a reduced consumption favorable to the process of sustainability.

At the level of Romania, the levels of the Transaction indicators, although they are significantly lower than MaxEU (by 4.9 points for Organic Farming and by 3.2 points for Genuine Savings) and closer to MinEU (with 3.2 points for the first indicator, respectively, and with 3.5 points to the second) however, by their values which exceed the central level, they indicate the positive trajectory in the development of the sustainable society.

A fairly significant impact on sustainability is presented by the two indicators of the Economy (GDP and Employment), a fact confirmed by the difference of 1.4 points compared to MaxEU. Public Debt represents the indicator with negative influence both at the European level (MaxEU being 9.7 points) and for Romania (8.5 points), and the high values, close to 10 points, indicate the high level of Public Debt with a negative impact on the sustainable society.

5.3.2. The Level of Human Well-Being

From the point of view of the Human Well-being dimension, the SSI values of the indicators included in this dimension (Table 14) highlight a state much higher than the world average in the Basic Needs category where all three values of the Sufficient Food, Sufficient Drinking Water and Safe Sanitation indicators are higher than the world average with values between 0.5 and 0.9 points.

**Table 14.** The extreme values of the indicators of the Human Well-being dimension compared to the values recorded worldwide.

| Category | Basic Needs | | | Personal Development | | | Well-Balanced Society | | |
|---|---|---|---|---|---|---|---|---|---|
| Indicator | Sufficient Food | Sufficient Drinking Water | Safe Sanitation | Education | Healthy Life | Gender Equality | Income Distribution | Population Growth | Good Governance |
| MaxUE | 10 | 10 | 10 | 10 | 10 | 8.4 | 7.9 | 10 | 8.7 |
| Romania | 10 | 10 | 8.7 | 9.0 | 9.3 | 7.4 | 7.0 | 9.1 | 5.8 |
| MinUE | 9.9 | 9.8 | 8.7 | 9.0 | 9.2 | 7.1 | 6.6 | 1.0 | 5.8 |
| World | 9.4 | 9.0 | 7.8 | 9.4 | 8.8 | 7.1 | 6.5 | 5.1 | 5.1 |

Additionally, in the EU27 framework, very good results are recorded, compared to the world average, regarding life expectancy at birth (the MinEU value of the Healthy Life indicator is higher than the world average by 0.4 points), the equitable distribution of income (the MinEU value of the Income Distribution indicator is higher than the world average by 0.1 points) and Good Governance (the MinEU value of the Good Governance indicator is higher than the world average by 0.7 points).

On the other hand, although the MaxEU value of the Education indicator is 10, its MinEU value, also recorded in Romania, is lower than the world average by 0.4 points, which represents a weak point. Moreover, a possible weak point is the control of population growth. The MinEU value of the Population Growth indicator is significantly lower than the world average, signifying the existence of states with an important population growth rate (with values significantly lower than 8).

As far as Romania is concerned, the value of the Population Growth indicator is 9.1 > 8, highlighting the existence of a population reduction process (negative demographic growth).

From the point of view of the Basic Needs category (Figure 14), Romania and Bulgaria, with 9.5 points, are among the last places in the ranking of EU countries.

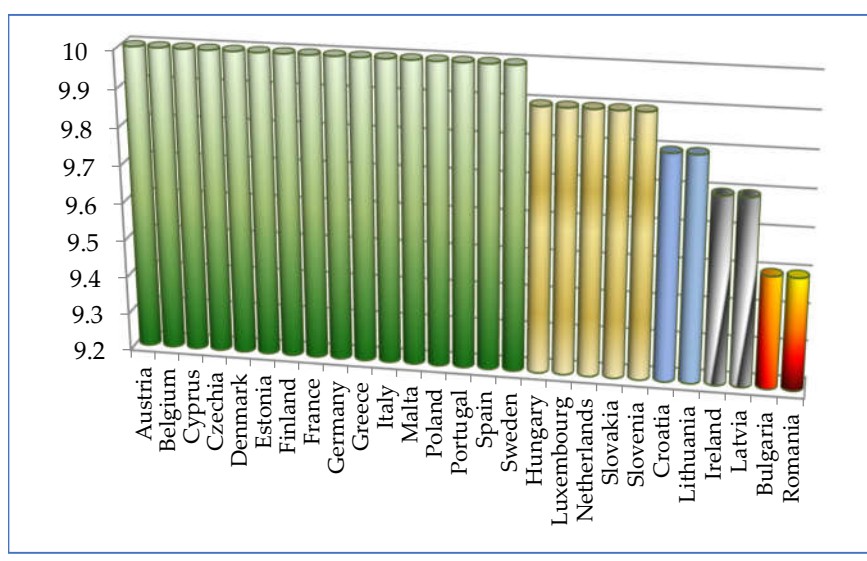

**Figure 14.** Hierarchy of the EU27 states from the point of view of the Basic Needs category.

Ireland and Latvia, with 9.7 points, respectively, and Croatia and Lithuania, with 9.8 points, represent four other countries in the ranking of EU countries that correspond to fairly low values of the points that highlight the level of Basic Needs.

A high score of 9.9 points belongs to five other EU countries (Hungary, Luxembourg, The Netherlands, Slovakia, and Slovenia). Therefore, the rest of the countries correspond to the maximum score of 10 points.

Regarding the Personal Development and Health category, the ranking of the countries is different from that of the previous category, with the scores forming seven groups (Figure 15). The amplitude of the ranking of 0.9 points is given by Finland and Sweden, with 9.4 points, placed at the top of the ranking, and Romania, with 8.5 points, occupying the last place.

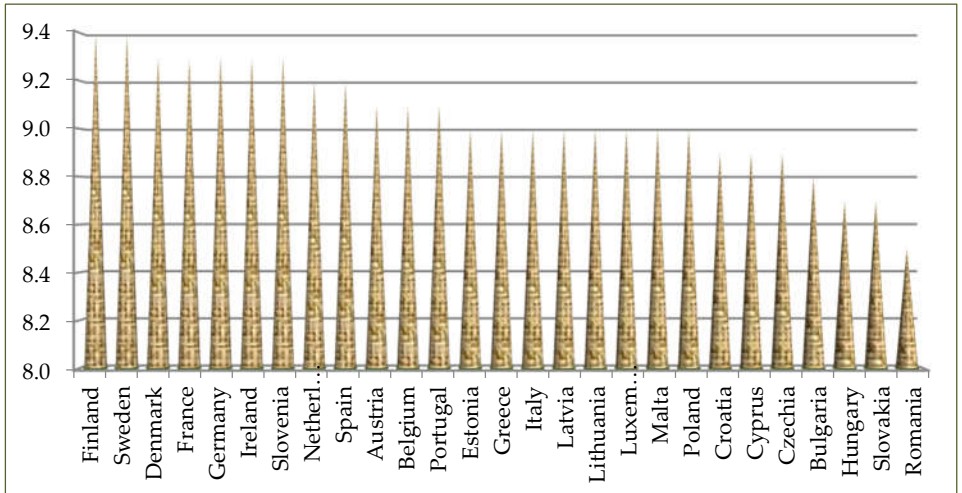

**Figure 15.** Ranking of the EU27 states from the point of view of the Personal Development and Health category.

The group formed by Denmark, France, Germany, Ireland, and Slovenia with 9.3 points is followed by that of the countries with 9.2 points (The Netherlands and Spain), to which is added the group corresponding to 9.1 points: Austria, Belgium and Portugal. If Hungary and Slovakia only have 8.7 points, and the group which includes Croatia, Cyprus, the Czech Republic and Bulgaria has 8.9 points per country, as can be seen from Figure 15, most EU countries (8 member states EU) correspond to 9 points.

The analysis carried out on the basis of the Well-balanced Society indicators brings to the forefront the manifestation of significant fluctuations in the scores recorded at the level of the EU countries, a fact noted by the formation of 13 groups of countries. The length of the score variation interval is also significant, 4.1 points, compared to that of the previous categories. This amplitude is determined in relation to the maximum of 7.9 points recorded as in the previous indicator also by Finland and the minimum of 3.8 points that correspond to Malta.

The last position among the EU member states occupied by Romania in the Personal Development and Health dimension can be explained on the one hand by the average life expectancy far below the European average (75.3 years compared to 81 years) [73], the accentuation of the aging phenomenon of population and the massive emigration of the young population, and on the other hand, the depreciation of education as a result of school dropouts and the lack of investment in school infrastructure.

From the hierarchy of the EU27 states from the point of view of the Well-balanced Society category (Figure 16), there are five groups with only one country in the ranking: Lithuania with 7.8 points, Sweden with 7 points, Ireland with 6.9 points, Cyprus with 6.8 points and Luxembourg with 5 points.

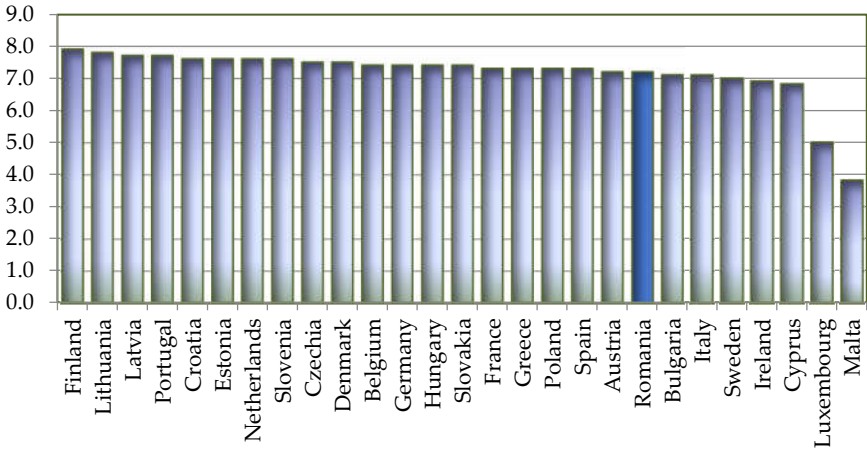

**Figure 16.** Hierarchy of the EU27 states from the point of view of the Well-balanced Society category.

Four other groups have two countries each: Latvia and Portugal with 7.7 points, the Czech Republic and Denmark with 7.5 points, Austria and Romania with 7.2 points, and Bulgaria and Italy with 7.1 points. Four countries each form three other groups, as follows: 7.6 points belong to the group formed by Croatia, Estonia, The Netherlands, and Slovenia; 7.4 points belong to the group with Belgium, Germany, Hungary, and Slovakia; and 7.3 points to the group with France, Greece, Poland, and Spain.

5.3.3. The Level of Environmental Well-Being

From the point of view of the Average Well-being dimension, both positive and negative results (weak points) are registered within the EU27.

Significant positive results recorded in the implementation of the sustainable society in the EU27, compared to the world average (Table 15), were recorded for the Biodiversity indicators (the MinEU value of 6.1 points is higher than the world level by 0.7 points), Consumption, where the necessary ecological footprint per inhabitant is significantly lower than worldwide, and Energy use, where the MaxEU value of 6.7 is lower than the global average by 0.1 points.

**Table 15.** The extreme values of the indicators of the Environmental Well-being dimension compared to the values registered at the world level.

| Category | Natural Resources | | | | | Climate and Energy | |
|---|---|---|---|---|---|---|---|
| **Indicator** | **Sufficient Food** | **Sufficient Drinking Water** | **Safe Sanitation** | **Energy Use** | **Education** | **Healthy Life** | **Gender Equality** |
| MaxUE | 9.0 | 9.9 | 1.0 | 6.7 | 7.3 | 10 | 5.7 |
| Romania | 8.6 | 9.7 | 1.0 | 6.7 | 3.1 | 6.3 | 3.1 |
| MinUE | 6.1 | 1.5 | 1.0 | 1.0 | 1.0 | 1.0 | 1.7 |
| World | 5.2 | 7.7 | 3.5 | 6.8 | 3.7 | 6.1 | 3.6 |

Some minuses compared to the world level are recorded in the indicators: Renewable Water Resources where MinEU = 1.5 (value registered in Malta) indicates a Renewable Water Resources level much lower than the world level), Energy Savings where MaxEU = 7.3 (value registered in Sweden) indicates increased energy consumption, while worldwide the value of 3.7 indicates a reduction, as well as regarding the level of greenhouse gas emissions and the share of renewable energy in total energy consumption.

From the point of view of the Environmental Well-being dimension, positive results were recorded in Romania in the Natural Resources dimension, while in the Climate and Energy dimension, some minuses were recorded in Greenhouse Gases with a greenhouse

effect, with the value of this indicator higher by 0.2 points compared to the value registered worldwide, as well as the share of renewable energy use, where the value of the Renewable Energy indicator (3.1 points) is lower than that corresponding to the global level.

From the point of view of the Natural Resources category (Figure 17), the score allowed for ranking the countries into eight groups.

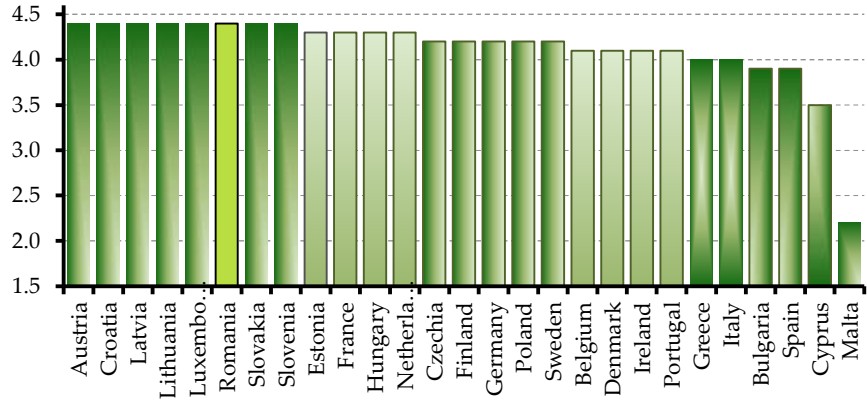

**Figure 17.** Hierarchy of the EU27 states from the point of view of the Natural Resources category.

Thus, with 2.2 points, Malta occupies the last place, previously Cyprus, with 3.5 points. There are two groups, each formed by two countries with 3.9 points (Bulgaria and Spain), respectively with 4 points (Greece and Italy). Next, it can be mentioned that there are two groups with four component countries each: with 4.1 points is the group formed by Belgium, Denmark, Ireland, and Portugal and with 4.3 points is the group that includes Estonia, France, Hungary, and The Netherlands. The value of 4.2 points belongs to the group that consists of five countries: Czech Republic, Finland, Germany, Poland, and Sweden. The group with the most countries (eight countries), which also includes Romania, has 4.4 points.

The ranking of the EU27 states from the point of view of the Climate and Energy category is more special (Figure 18) in the sense that 18 groups are formed in conditions where there are many differences from one group to another, the values are very small, and the amplitude is quite significant (3.1 points).

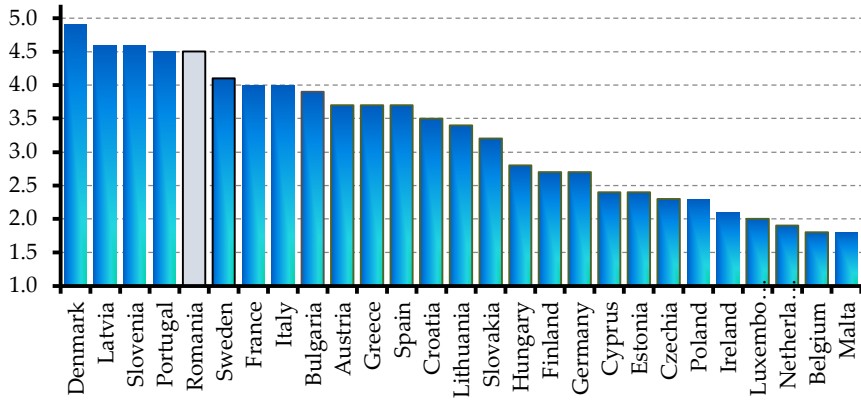

**Figure 18.** Hierarchy of the EU27 states from the point of view of the Climate and Energy category.

The country ranking includes 10 groups consisting of only one country: Denmark at the top of the ranking with 4.9 points, followed by Sweden with 4.1 points, Bulgaria with 3.9 points, Croatia with 3.5 points, Lithuania 3.4 points, Slovakia with 3.2 points, Hungary with 2.8 points, Ireland with 2.1 points, Luxembourg with 2 points and The Netherlands with 1.9 points. The ranking continues with 7 groups which include two countries each: Latvia and Slovenia with 4.6 points each, Portugal and Romania with 4.5 points each, France and Italy with 4 points each, Finland and Germany with 2.7 points each, Cyprus and Estonia with 2.4 points each, the Czech Republic and Poland with 2.3 points each, and Belgium and Malta with 1.8 points each. In the ranking there is only one group with three countries, Austria, Greece and Spain, which correspond to 3.7 points each.

### 5.3.4. The Level of Economic Well-Being

The third dimension of the SSI includes the Transition and Economy categories. Among the indicators that are part of these categories, the EU27 as a whole is characterized by higher results than the world level for the GDP indicator, with the corresponding MinEU value (7.9 points) being 2.4 points higher than the world level. In the case of the other two indicators of the Economy category, both the Employment rate and the share of Public Debt differ significantly from one state to another, with the differences between the MaxEU and MinEU values being very large (Table 16).

A similar situation is also registered in the case of the Organic Farming and Genuine Savings indicators, where, due to the wider range of their values, there are countries where the level of transition towards a sustainable society is higher than the world level, and others where this level is lower.

**Table 16.** The extreme values of the indicators of the Economic Well-being dimension compared to the values recorded worldwide.

| Category | Natural Resources | | | Climate and Energy | |
|---|---|---|---|---|---|
| Indicator | Organic Farming | Genuine Savings | GDP | Employment | Public Debt |
| MaxUE | 10 | 9.2 | 10 | 8.0 | 9.7 |
| Romania | 5.1 | 6.0 | 8.6 | 6.6 | 8.5 |
| MinUE | 1.9 | 2.5 | 7.9 | 1.5 | 1.0 |
| World | 2.7 | 8.3 | 5.5 | 6 | 5.1 |

In the case of Romania, higher values were recorded than worldwide for the share of areas converted or in the process of being converted to Organic Farming, for GDP/inhabitant and for the degree of employment. Furthermore, considering that the score obtained for the Public Debt indicator is higher than the score recorded at the world level, it follows that the share of public debt in GDP is lower than that at the world level. The weak point is the score obtained for the Genuine Savings indicator, which means that the level of adjusted net savings in Romania is below the world level.

As in the case of the Climate and Energy category, a component of Environmental Well-being, the Transition category, which forms Economic Well-being, is characterized by a classification made up of many groups, one more than the one mentioned comparatively (Figure 19).

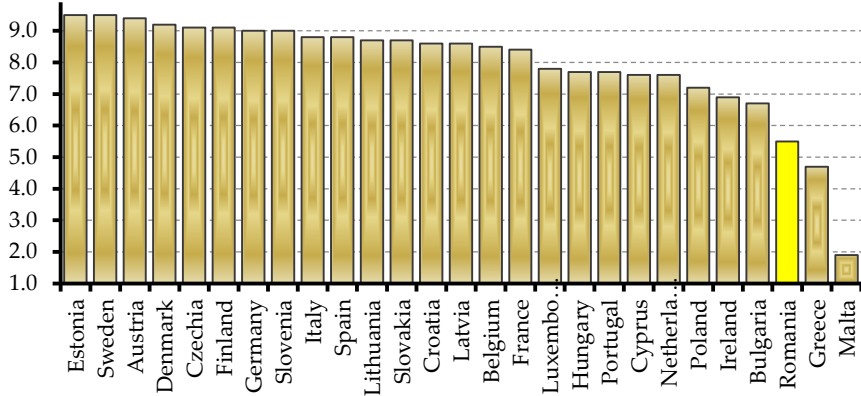

**Figure 19.** Hierarchy of the EU27 states from the point of view of the Transition category.

The particularity of the classification of this category consists in the fact that there are only two types of groups: some of which are composed of a single country and others formed by only two countries. Thus, the first type of groups includes Austria (9.4 points), Denmark (9.2 points), Belgium (8.5 points), France (8.4 points), Luxembourg (7.8 points), Poland (7.2 points), Ireland (7.9 points), Bulgaria (6.7 points), Romania (5.5 points), Greece (4.7 points), Malta (1.9 points). The second type of group consists of: 9.5 points for the group with Estonia and Sweden (dominating the ranking); 9.1 points for the group with the Czech Republic and Finland; 9 points for the group containing Germany and Slovenia; then 8.8 points and 8.7 points for the group with Italy and Spain, and Lithuania and Slovakia, respectively; followed by 8.6 points belonging to the group with Croatia and Latvia; and 7.7 points and 7.6 points belonging to the bottom of the ranking of the last groups formed from Hungary and Portugal, and Cyprus and The Netherlands, respectively.

The scores recorded determined a ranking of the EU27 states from the point of view of the Economy category (the second component of Economic Well-being), in 20 groups (Figure 20). The groups formed are of three types: groups consisting of a single country, groups with two countries and groups consisting of three EU27 member states.

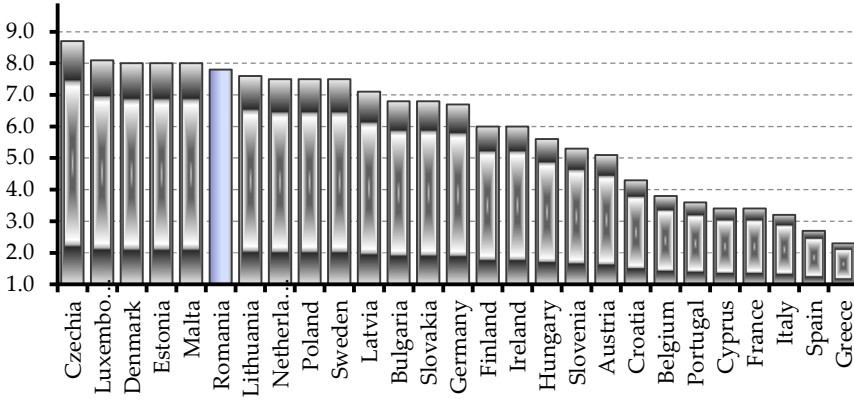

**Figure 20.** Hierarchy of the EU27 states from the point of view of the Economy category.

The analysis carried out for the Economy category based on the ranking of the EU27 member states takes into account the length of the variation interval of 6.4 points, the largest among all others which belong to the Human Well-being, the Environment and the Economic categories. The obtained amplitude is the result of the difference between the maximum score recorded by the Czech Republic of 8.7 points and the minimum of 2.3

points belonging to Greece. Thus, most states, in fact, three countries each, register 8 points (Denmark, Estonia, Malta) and 7.5 points (The Netherlands, Poland, Sweden). Three other groups with two countries each had 6.8 points (Bulgaria, Slovakia), 6 points (Finland, Ireland) and 3.4 points (Cyprus, France). The 15 groups, i.e., the other 15 countries, correspond to values that oscillate (excluding the maximum and the minimum) between 8.1 points recorded for Luxembourg and 2.7 points belonging to Spain.

## 6. Conclusions

In a global economic system, the balance between needs and resources in terms of ensuring a sustainable future must be permanently monitored in order to identify vulnerabilities and risks that can affect sustainability.

The SSI is a sustainability monitoring tool that allows comparative analysis of countries' progress and can be used in simulations and the strategic planning and decision-making process for a sustainable future.

The statistical analysis of SSI indicators on the European continent highlights the positive situation in the manifestation of sustainability at the level of Northern Europe, especially in indicators such as GDP, Organic Farming, Biodiversity and Good Governance.

In the implementation of the sustainable society at the EU27 level, it was demonstrated that the most important role was played by the following indicators of Human Well-being: Sufficient Food, Sufficient Drinking Water, Safe Sanitation, Healthy Life, Income Distribution and Good Governance. Education and Population Growth represented two other indicators of this Welfare category whose values highlighted an unfavorable situation in the direction of sustainable society development, requiring the adoption and application of appropriate strategic decision-making measures in order to stimulate them.

A favorable situation in the development of the sustainable society is reflected mainly by the SSI values, which were higher than the world average, for two indicators of Environmental Well-being: Biodiversity and Consumption. Within this category, Renewable Water Resources, Energy Savings, Greenhouse Gases and Renewable Energy have an unfavorable impact on the sustainable society as a priority, which must be acted upon to stimulate the factors which will lead to positive results in the future.

At the same time, the analysis highlighted that most indicators of Economic Well-being (less Public Debt) are significant in the positive development of a sustainable society.

As far as Romania is concerned, the deficient indicators for the development of a sustainable society, on which action must be taken, are Population Growth as a component of Human Well-being, Climate and Energy through Greenhouse Gases and Renewable Energy that form Environmental Well-being, and respectively the Real Economy and Debt Public as components of Economic Well-being.

By 2050, the European Union, by adopting the Climate Law [74], has set itself a major objective, namely, to become neutral from a climate point of view, which is in accordance with the previously adopted strategic document European Green Deal [75].

In order to achieve the objectives, which aim to mitigate climate change and sustainable well-being, public information and educational policies that focus on the links between social and environmental issues [76] and partnerships for objectives [77], are of particular importance.

The educational factor should be the catalyst for the changes that are expected and are applicable through SSI to the transformation of the entire society in the direction of sustainable development. Thus, the modeling of the educational level aimed at sustainable development must be oriented towards raising the level of awareness of each individual.

Achieving a high level of sustainability requires, in the long term, the application of specific development policies by the activity sector in relation to the trends of each indicator aimed at raising the level of sustainability highlighted by the SSI. The implementa-

tion of policies opens the perspective of adequately implementing sustainable development plans in the direction of maintaining a balance between the three components of sustainability: economic, social and ecological, in order to satisfy both the current and future needs of the population.

In the analysis carried out, there are limitations related to the difficulties that arise, mainly in the collection of data, but also in the design, completion or rethinking of the system of indicators that form the SSI, taking into account the limitation of the notion of sustainability. At the same time, it is necessary to take into account the limitations that appear even in the definition and the degree of commensurability of some indicators, which, over time, would allow their replacement with others that reflect the sustainability process much better. This would represent not only a limitation but also an opportunity for future research in the sense of identifying new indicators that better reflect the level of SSI.

Research in the future can be completed with a comparative evaluation of the strategic actions of involvement at the national level to show the trajectory traveled by each European state in the process of recovery and resilience, thus being able to identify models of sustainable mobility.

**Author Contributions:** Conceptualization, R.-M.G. and M.Z.; methodology, M.Z.; software, M.Z.; validation, A.B., R.-M.G. and M.Z.; formal analysis, A.B., R.-M.G. and M.Z.; investigation, A.B, R.-M.G. and M.Z.; resources, G.A.C.; data curation, R.-M.G.; writing—original draft preparation, A.B.; writing—review and editing, G.A.C.; visualization, G.A.C.; supervision, A.B.; project administration, R.-M.G.; funding acquisition, G.A.C. All authors have read and agreed to the published version of the manuscript.

**Funding:** This research received no external funding.

**Institutional Review Board Statement:** Not applicable.

**Informed Consent Statement:** Not applicable.

**Data Availability Statement:** Not applicable.

**Conflicts of Interest:** The authors declare no conflict of interest.

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
