# Peer review of "The Image of Sustainability in European Regions Considering the Social Sustainability Index"

_sustainability, doi:10.3390/su142013433_

Round 1

Reviewer 1 Report

Review report on the image of sustainability in European regions considering  the Social Sustainability Index

I have gone through the manuscript and found it fit for the scope of this journal. The issues researched are relevant and stand the potential of advancing knowledge in the extant literature. Specifically, an empirical research on sustainability at this period is highly demanding for saving the global economy from extinction. I give kudos to the authors for bringing this topic into the limelight. That being said, I provide the following comments for the authors to consider.

1.     On the abstract:

a.      Authors should provide some motivational statements about at least one or two contributions of the study in the abstract.

b.      The method of analysis which I believed is descriptive should be clearly stated.   

c.      One or two policy insights from the study should be stated.

2.     On the introduction: The introductory section is fine but could be improved. I provide a few points.

a.      Authors should strengthen the motivations of the study and provide more arguments on contribution of the study. For instance, how does this study differs from previous studies? Besides, since various studies have been conducted on sustainable development using varying indicators, these indicators should be enlisted and arguments should be provided on why the indicator adopted by the current study is better.

b.     I think it will be very difficult to discuss sustainability in the present moment without making reference to the recent United Nations Climate Change Conference held in 2021 (COP26).

c.      There should be a paragraph at the end of the introduction on the structure of the work.

3.     On literature review

a.      This section is totally missing. Was it intentional?

b.     There is need for authors to review certain studies relating to the subject matter. Without reviewing what are existing, lacunas cannot be identified and contributions will be difficult to advance.

c.      I suggest that authors create a section to review studies at least from 2018 to 2022 dealing with sustainability from diverse country groups across the globe. The following studies could be useful: https://doi.org/10.3390/su14084415; 10.1353/jda.2021.0022;

d.     Authors should create one paragraph at the end of section to appraise the studies reviewed and position the contributions of the current study through the lacunas in the extant studies.

4.     The discussion of results should be linked to recent studies.

5.     Conclusion can be improved.

6.     The policy recommendations are weak in their present form. They can strengthen it by relating the findings to the recommends.

7.     Limitation and future research opportunity should be identified. 

Overall, the paper is full of potentials that can be evident if the comments above are addressed.

Reviewer 2 Report

I would encourage the authors to receive these comments as a constructive way to improve and strengthen their paper.

The paper doesn`t have a strong theoretical framework, it does not provide sufficient background information and literature review regarding the topic of Social Sustainability Index (SSI). The literature review should be reinforced with up-to-date and focused bibliography on the particular theme of SSI. The authors can augment and improve this area of the manuscript before they present the materials and methods.

Although this is an application, by comparison with SSI to the case of Romania, the approach has a brief overall framework to apply to a particular country in this matter. It would be interesting to mention if there are other similar applications in other countries to make this comparison.

There are other distinct influential variables in the country (geographical; social; cultural; or political) that when explored in the theoretical part will certainly be emphasized in the empirical part and may influence Romania's performances. These variables should be highlighted in the discussion.

The application of the model is limited, we are comparing a global model to a particular case so all the slices of the model (index) should be carefully explored for the country in question. The results cannot be generalized to other sustainable issues and are restrictive for a local application in a country that has a distinct governmental policy and social characteristics different from the other European economies. 

Following this procedure will make it easy to complement the discussion of the results.

Reviewer 3 Report

Dear Authors,

My first thoughts when I read the article were to suggest Reject. But I'd like to see it go ahead and get published. However, it needs many changes to move forward, so I suggested a Major revision. Below I have tried 5 points to give you just the main points to correct and understand what you have done and what you need to change.

1. There is no Methodology section. So, this article is not an original article. It is a Review article.

In my opinion, you should redefine the paper's purpose and clarify the added value given to the academic community. 

2. Where can these graphs be used?

3. You talk about regions. So, why have you not used maps (for example, GIS)? Without spatial indications, there is no reason to have such an article.

4. There are entire paragraphs without references. Please recheck the manuscript carefully.

5. You need to add more targeted keywords

There is no way that an article will be published without those changes.

 I hope my comments help you.

Round 2

Reviewer 1 Report

The authors have done considerable justice to the issues raised and earned their work the merit of acceptance. Hence, I recommend the work for acceptance in its present state. Congratulations and kudos to the authors for a well done job.

Author Response

Dear Sir/Madam

Thank you for giving us the opportunity to publish our manuscript to Sustainability.

We appreciate the time and effort that you have dedicated to providing your valuable feedback on our manuscript.

Compared to the last version of the article, we have been able to incorporate small additions to reflect the suggestions provided by the Reviewer 3 (1. adding software for maps - see the last paragraph of methodology and 2. all spelling and grammatical errors have been corrected).  We have highlighted the changes in the manuscript using yellow text.

Reviewer 2 Report

I am satisfied with the corrections and improvements in the article and agree with the version now presented.

Author Response

(The authors gave the same response as above.)

Reviewer 3 Report

Dear Authors,

I am delighted with your changes. I will recommend two more things for your article.

1. You must write down which program you used to export the maps. Also, the images should be in a better resolution.

2. You need to add more References. Look carefully at the article one more time and get it right. 

These are critical issues and should be reported and improved. Never forget them!

Author Response

Dear Sir/Madam,

We appreciate the time and effort that you have dedicated to providing your valuable feedback on our manuscript.

We have added the software used for maps, as you suggested (see the last paragraph of methodology)

Also, we have added 2 references and corrected the spelling and grammar of the text.

We have highlighted the changes in the manuscript using yellow text.
